# The Bioactivity Prediction of Peptides from Tuna Skin Collagen Using Integrated Method Combining In Vitro and In Silico

**DOI:** 10.3390/foods10112739

**Published:** 2021-11-09

**Authors:** Liza Devita, Hanifah Nuryani Lioe, Mala Nurilmala, Maggy T. Suhartono

**Affiliations:** 1Department of Food Science and Technology, Faculty of Agricultural Engineering and Technology, Bogor Agricultural University, Bogor 16680, Indonesia; liza.devita1981@gmail.com (L.D.); hanifahlioe@apps.ipb.ac.id (H.N.L.); 2The Ministry of Agriculture Republic Indonesia, Jakarta 12550, Indonesia; 3Department of Aquatic Product Technology, Faculty of Fisheries and Marine Sciences, Bogor Agricultural University, Bogor 16680, Indonesia; mnurilmala@apps.ipb.ac.id

**Keywords:** *Thunnus obesus*, fishery by-product, collagen protein, enzymatic hydrolysis, bioactive peptides, ultrafiltration peptide fractions, antioxidant activity, in silico

## Abstract

The hydrolysates and peptide fractions of bigeye tuna (*Thunnus obesus*) skin collagen have been successfully studied. The hydrolysates (HPA, HPN, HPS, HBA, HBN, HBS) were the result of the hydrolysis of collagen using alcalase, neutrase, and savinase. The peptide fractions (PPA, PPN, PPS, PBA, PBN, PBS) were the fractions obtained following ultrafiltration of the hydrolysates. The antioxidant activities of the hydrolysates and peptide fractions were studied using the DPPH method. The effects of collagen types, enzymes, and molecular sizes on the antioxidant activities were analyzed using profile plots analysis. The amino acid sequences of the peptides in the fraction with the highest antioxidant activity were analyzed using LC-MS/MS. Finally, their bioactivity and characteristics were studied using in silico analysis. The hydrolysates and peptide fractions provided antioxidant activity (6.17–135.40 µmol AAE/g protein). The lower molecular weight fraction had higher antioxidant activity. Collagen from pepsin treatment produced higher activity than that of bromelain treatment. The fraction from collagen hydrolysates by savinase treatment had the highest activity compared to neutrase and alcalase treatments. The peptides in the PBN and PPS fractions of <3 kDa had antidiabetic, antihypertensive and antioxidant activities. In conclusion, they have the potential to be used in food and health applications.

## 1. Introduction

Bioactive peptides are protein fragments that provide nutrition and positive benefits for body functions [1]. An array of bioactive properties of peptides and proteins can be seen in Figure 1. These peptides can function as antidiabetic, antihypertensive, antimicrobial, antioxidant and others [2,3,4]. Their functionalities are influenced by the type, composition, and amino acid sequence of the peptides [5]. Bioactive peptides can have multifunctional properties, namely peptides that have more than one bioactive property [6,7,8].

Various proteins can act as sources of bioactive peptides [17,18,19,20,21]. One of the most recent studies is the study of bioactive peptides from collagen by-products of the fisheries industry. Several studies of bioactive peptides from collagen by-products in the fisheries industry are as follows. Antioxidant activities have been studied for the hydrolysates/peptides: salmon scales [22], skipjack tuna bone [23], seabass skin [24], Spanish mackerel skin [25,26], milkfish scales [27], tilapia scales [28], and tilapia skin [29]. In these studies, antioxidant activities were studied using a scavenging activity test against superoxide anion, DPPH (2,2-diphenyl-1-picrylhydrazyl) radical, ABTS (2,2′-azino-bis-(3-ethylbenzothiazoline-6-sulfonic acid)) radical, and OH radical, FRAP (ferric reducing antioxidant power) or reducing power assays, and ferrous or metal chelating activity. Bioactive peptides of the peptides/hydrolysates from collagen by-products of the fisheries industry have also been studied for their anti-inflammatory activity [27], hypoglycemic effect [29], and antiaging [30].

Research on bioactive peptides from collagen by-products of the fisheries industry is important for the following reasons. First, because it produces bioactive compounds that have the potential to be utilized in food, pharmaceuticals, or cosmetic products. Second, because it can provide added value for the fisheries industry. Third, because it is a form of concern in managing environmental impacts that may arise from the by-products of the fisheries industry.

Collagen peptides can be prepared by a two-step hydrolysis process of the by-products of the fisheries industry [31]. Hydrolysis can be carried out enzymatically using proteases, and the collagen peptides can then be studied for their bioactivity [32,33].

The study of the bioactive properties of a peptide can be carried out either in vitro, in vivo, and in silico. Research on peptides in vitro and in vivo can be quite expensive, requires a long time and is painstaking work. The use of web-based in silico methods is able to facilitate the research of bioactive peptides [34,35,36,37]. The combination of methods (integrated approach) between in vitro, in vivo, and in silico methods can be carried out. Good correlation between experimental analysis (in vitro or in vivo) and in silico needs to be built to effectively utilize the integrated approach in finding bioactive peptides from various proteins [36]. In vitro tests can be carried out as an initial study to find one or more bioactive properties of peptides. In vivo testing is needed to see how peptides that have been studied can benefit the body. In silico studies can be an alternative to in vitro and in vivo studies to find the bioactive properties. Some studies have dealt with in vitro and in silico to obtain information on bioactive peptides. Even though in vitro and in vivo studies have been conducted to study antimicrobial activity, and small antimicrobial peptides could be explored [38], in silico and in vitro studies could also be used to confirm the multifunctionality of peptides [39].

Tuna is classified as a fish in high demand worldwide. Indonesia is one of the largest tuna-producing countries in the world [40]. Tuna is one of Indonesia’s main export commodities [41]. The economic value of Indonesian tuna fishery products is an opportunity that can continue to be exploited. Indonesia’s tuna export volume in 2017 reached 198,131 tons [42]. Tuna is usually consumed as sashimi or other types of dishes and canned food. The tuna fishery industry only utilizes certain parts of the fish, thus producing by-products (including skin). One of the tuna species is the bigeye tuna used in this study. These by-products can be processed as animal feed and crackers and also processed into collagen and enzymatically hydrolyzed to obtain bioactive peptides [32,33,43,44,45,46].

Comprehensive studies on bioactive peptides from collagen by-products of the tuna fishery industry, especially from collagen of the skin of tuna bigeye (*Thunnus obesus*), are still limited. Therefore, the objectives of this study were to study the hydrolysates and peptide fractions of bigeye tuna skin collagen as antioxidant compounds; to explore the effects of several factors (collagen types, enzymes, and molecular sizes) on the antioxidant activities produced; to analyze the amino acid sequences of the peptides in the fractions with the highest antioxidant activity; and to explore the bioactivities and characteristics of the peptides in the fractions with the highest antioxidant activity by in silico studies.

## 2. Materials and Methods

### 2.1. Materials

The skin of tuna (bigeye tuna, *Thunnus obesus*) was obtained from PT Maluku Prima Makmur, Ambon, Indonesia. Other materials were 1-butanol, sodium hydroxide, acetic acid (glacial) 100%, pepsin (700 FIP-U/g, 0.7 Ph Eur-E/mg; EC 3.4.23.1), sodium chloride GR for analysis, ethanol absolute for analysis, methanol gradient grade for liquid chromatography, L(+)-ascorbic acid, buffer solution (di-sodium hydrogen phosphate) pH 7.00 (20 °C), purchased from Merck (Darmstadt, Germany); bromelain from pineapple stem (≥3 units/mg protein; EC 3.4.22.32), alcalase or protease from *Bacillus licheniformis* (≥2.4 U/g), 2,2-diphenyl-1-picrylhydrazil, bovine serum albumin, purchased from Sigma-Aldrich (St. Louis, MO, USA); neutrase 0.8 L (0.8 AU-N/g), savinase 16 L (16 KNPU-S/g), purchased from novozymes A/S (Denmark, Germany); distilled water, purchased from IKA Pharmindo Putramas Co. (Jakarta, Indonesia).

The explanation of the activity unit of the protease enzyme is as follows: 1 unit (U) is the amount of enzyme that catalyzes the reaction of 1 µmol of substrate per minute. One Novo Protease Units (NPU) is the amount of enzyme that hydrolyzes casein at such a rate that the initial rate of formation of peptides/minute corresponds to 1 micromole of glycine/minute, meanwhile 1 KNPU (Kilo NPU) equals 1000 NPU. One Anson unit (AU) is approximately equal to 3 KNPU. Baseline lipase, amylase, and free and total protease activity levels were determined for each sample in accordance with procedures specified by the Fédération Internationale Pharmaceutique (FIP) and the European Pharmacopoeia (PhEur). Results were converted from FIP/PhEur units to United States Pharmacopeia (USP) units. For protease, 1 FIP/PhEur unit = 62.5 USP units.

### 2.2. Skin Preparation and Collagen Extraction

The skin was prepared according to the method [47]. The skin used was bigeye tuna (*Thunnus obesus*) skin, which was a fishery by-product. The skin was from Ambon and brought to the research laboratory in Bogor. The skin was stored in the freezer (−28 °C). The skin was cleaned, cut into small pieces (0.5 × 0.5 cm^2^), and then lyophilized for 14 h using a TFD5503 Bench-Top freeze dryer (ilShinBioBase, Ede, Netherlands). The lyophilized skin was used for collagen extraction.

Collagen extraction was carried out according to the method in [47]. The first stage was the pre-treatment stage, namely soaking of the skin in 0.1 M NaOH solution, followed by soaking in 10% (*v*/*v*) butyl alcohol solution. Soaking was carried out in the refrigerator at 4 °C for 24 h. The ratio of skin and soaking solution was 1:10 (*w*/*v*). The soaking solution was changed every 12 h. At the second stage, the pre-treated skin was extracted using dilute acetic acid (0.5 M) containing 0.1% protease (*w*/*v*). The proteases used were bromelain and pepsin. Extraction was carried out at 4 °C (in the refrigerator) for 72 h. The ratio of pre-treated skin to extraction solution was 1:40 (*w*/*v*). During the pre-treatment and extraction processes, the solution must be shaken frequently. The suspension was then centrifuged using a refrigerator centrifuge Hermle Z 32 HK (Benchmark Scientific, Sayrevill, NJ, USA) at 6000× *g* for 30 min at 4 °C. The supernatant was separated from the precipitate. The supernatant was then salted-out by sodium chloride to a final concentration of 2.0 M. The precipitate was collected by centrifugation at 6000× *g* for 30 min at 4 °C and then dialyzed with cold distilled water using a dialysis membrane, Spectra/Por^®^ Dialysis Membrane MWCO 6–8000, Regenerated Cellulose (Spectrum Laboratories, Inc., Rancho Dominguez, Los Angeles, CA, USA). The obtained wet collagen was then lyophilized for 14 h using TFD5503 Bench-Top freeze dryer (ilShinBioBase, Ede, The Netherlands), so that dry collagen (freeze dry collagen) was obtained. Collagen obtained from skin extraction using acetic acid and bromelain was called bromelain soluble collagen (BSC). The collagen obtained from skin extraction using acetic acid and pepsin was called pepsin soluble collagen (PSC).

### 2.3. Preparation of Hydrolysates and Peptide Fractions

Hydrolysates and peptide fractions were prepared according to the method in [48]. The hydrolysates were the results of hydrolysis of collagens (bromelain soluble collagen and pepsin soluble collagen) using alcalase, neutrase and savinase. The peptide fractions were the results of semi-purification of the hydrolysates by an ultrafiltration technique using 10 and 3 kDa tubes (Amicon^®^ Ultra-15 Centrifugal Filters Ultracel^®^-3K and 10 K, regenerated cellulose) (Merck Millipore Ltd. Tullagreen, Carrigtwohill, Co. Cork, Ireland).

Hydrolysate preparation was carried out as follows. A total of 200 mg of freeze dry collagen was dissolved in 10 mL of solution A (20 mg of an enzyme in 50 mL of phosphate buffer pH 7) in a 50 mL covered Erlenmeyer. The mixture was homogenized and hydrolyzed at pH 7, temperature 50 °C for 6 h in a SW22 Shaking Water Bath (JULABO GmbH, Seelbach, Germany). A shaking water bath was used to obtain the homogeneity condition of the mixture and temperature. Then, the enzyme was inactivated. Enzyme inactivation was carried out in boiling water for 10 min. The mixture was cooled, then centrifuged for 30 min at 6000× *g* in a Hermle Z 32 HK (Benchmark Scientific, Sayrevill, NJ, USA). The resulting supernatant was a hydrolysate. Three hydrolysates were produced from each collagen.

The preparation of peptide fraction was as follows. A total of 10 mL of the hydrolysate was placed into a 10 kDa ultrafiltration tube, then centrifuged by a refrigerator centrifuge Hermle Z 32 HK (Benchmark Scientific, Sayrevill, NJ, USA) at 6000× *g* for 30 min. Two peptide fractions were obtained, which were sized above and below 10 kDa. Then, the peptide fraction <10 kDa was replaced into a 3 kDa ultrafiltration tube. The filtration process using a 3 kDa ultrafiltration tube was carried out in the same way as the 10 kDa tube ultrafiltration process. In the ultrafiltration process with a 3 kDa tube, two peptide fractions were obtained with the molecular sizes 3–10 and <3 kDa. Thus, each hydrolysate produced three peptide fractions.

### 2.4. Protein Concentration Measurement of Hydrolysates and Peptide Fractions

Analysis of protein concentrations for hydrolysates and peptide fractions were carried out using the method in [49]. The absorbances were read at 500 nm using a Jenway 7315 Spectrophotometer (Bibby Scientific Ltd., Staffordshire, UK). The protein concentrations of the hydrolysates and peptide fractions were obtained by plotting the unknown concentrations on a standard curve of bovine serum albumin (BSA).

### 2.5. Antioxidant Activity Test

The antioxidant activities of the hydrolysates and peptide fractions were determined using the scavenging method against DPPH radicals according to the method [50]. Each hydrolysate was mixed with ethanol (1 mg mL^−1^). Approximately 100 µL of the solutions were added with 100 µL of DPPH (125 µM in ethanol). The mixtures were homogenized and incubated for 30 min at room temperature and protected from light. Ascorbic acid was used as a positive control. The absorbances were determined at 517 nm using a microplate reader (Epoch Microplate Spectrophotometer, BioTek Instruments, Inc., Winooski, VT, USA). The same procedures were conducted to measure the antioxidant activities of the peptide fractions. Antioxidant activities for the hydrolysates and peptide fractions were reported in µmol AAE/g protein.

### 2.6. Statistical Study of the Effects of Collagen Types, Enzymes, and Molecular Sizes on Antioxidant Activities

The antioxidant activities obtained at point 2.4 were then also studied statistically analyzed using the analysis of profile plots in the SPSS program. These analyses were conducted to see the influence of several factors on the antioxidant activities produced. The factors studied were collagen types (bromelain soluble collagen, pepsin soluble collagen), collagen hydrolyzing enzymes (alcalase, neutrase, and savinase), and molecular sizes (peptide fraction >10 kDa, peptide fraction 3–10 kDa, and peptide fraction <3 kDa). The procedure for analyzing profile plots was as follows: (1) opened *the SPSS program*, (2) input data in *the variable view*, (3) set *the data view*, (4) took *multivariate profile plots*, (5) clicked *continue*, (6) clicked *plot*, set *the dependent variables* and *the fixed factors*, (7) clicked *ok*, (8) saved, and (8) opened *the SPSS output*.

### 2.7. Amino Acid Sequence Analysis Using LC-MS-MS

The PBN and PPS fractions <3 kDa were compounds that had the highest antioxidant activity of all the tested compounds using the scavenging method against DPPH radicals at point 2.4. Amino acid sequence analysis was performed on the PBN and PPS fractions <3 kDa. The amino acid sequences of the peptides in the PBN and PPS fractions of <3 kDa were studied according to the method in [51]. The amino acid sequences of the peptides in the PBN and PPS fractions of <3 kDa were identified using UHPLC Vanquish Tandem Q Exactive Plus Orbitrap HRMS Thermo Scientific with column C18 (Pep Map RSLC C18 column, 150 × 0.075 mm, 3 µm, 100 Å) (Thermo Fisher Scientific Inc., Tewksbury, MA, USA) located at the Advanced Laboratory of the Bogor Agricultural University (Indonesia). Approximately 5 μL of the PBN and PPS fractions <3 kDa were inserted into the column. Eluent A and eluent B were used as the mobile phase. Eluent A was water containing 0.1% formic acid and eluent B was acetonitrile containing 0.1% formic acid. The linear gradients at flow rates of 0.3 µL/min and 30 °C were as follows: 0–3 min (2% B), 3–30 min (2–35% B), 30–45 min (35–90% B), 45–60 min (90% B), 60–60.1 min (90–5% B), 60.1–90 min (5% B). Instrument conditions were as follows: electrospray ionization source (ESI), positive mode, mass range 200–2000 m/z. Data analysis was performed using Proteome Discoverer 2.2 (Thermo Fisher Scientific Inc., Tewksbury, MA, USA).

### 2.8. In Silico Analysis for Bioactivity Prediction and Characteristics Study

#### 2.8.1. Bioactivity Prediction Using BIOPEP

Prediction of the bioactivities of peptides in the PBN and PPS fractions of <3 kDa was studied using a web-based in silico application that can be accessed on the BIOPEP website [52]. The analysis was carried out based on information obtained from the literature in [36].

#### 2.8.2. Characteristics of Peptides Using PepDraw

The study of the characteristics of bioactive peptides in the PBN and PPS fractions of <3 kDa was explored using a web-based in silico analysis that could be accessed on the PepDraw website [53]. The study of the characteristic of the bioactive peptides in the PBN and PPS fractions of <3 kDa was carried out by entering their amino acid sequence data, based on the information from the literature of [36].

### 2.9. Statistical Analysis

Data are presented as the mean ± standard deviation (SD). The data were conducted using Microsoft Excel 2016 (Microsoft Corp, Redmond, WA, USA) for the one-way ANOVA and IBM SPSS Statistics Version 20 for Windows (SPSS Inc., Chicago, IL, USA) for the profile plots analysis.

## 3. Results

### 3.1. Collagen Extraction

Collagens used for the preparation of hydrolysates and peptide fractions in this study were collagens obtained from the extraction process of bigeye tuna (*Thunnus obesus*) skin using dilute acetic acid (0.5 M) containing bromelain and pepsin. The collagen extracted using dilute acetic acid and bromelain was called bromelain soluble collagen (BSC). The collagen extracted using dilute acetic acid and pepsin was called pepsin soluble collagen (PSC). The isolation process was carried out in two stages, pre-treatment and collagen extraction. Pre-treatment aimed to remove the non-collagenous part, so that pure collagens were expected to be obtained. The yield and chemical characteristics of these collagens (BSC and PSC) have been described in our previous study [46], which also stated that the collagens (BSC and PSC) had high yields and had antioxidant activities, so they have the potential to be studied further.

### 3.2. Hydrolysates and Peptide Fractions

Hydrolysates of collagen were prepared from collagens (BSC and PSC) by hydrolyzing the collagens (BSC and PSC) using three types of proteases, namely alcalase, neutrase, and savinase. The hydrolysates were dissolved as the supernatant. HBA, HBN, and HBS were BSC hydrolysates, which were hydrolyzed by alcalase, neutrase, and savinase, respectively. And, HPA, HPN, and HPS were PSC hydrolysates, which were hydrolyzed by alcalase, neutrase, and savinase, respectively. Thus, there were six hydrolysates resulting from the hydrolysis of BSC and PSC using these three proteases. In other words, each collagen produces three hydrolysates.

The peptide fractions were obtained from the semi-purification process of hydrolysates using two 10 and 3 kDa ultrafiltration membranes. PBA, PBN, and PBS were the ultrafiltrated fractions of HBA, HBN, and HBS, respectively. PPA, PPN, and PPS were the ultrafiltrated fractions of HPA, HPN, and HPS, respectively. Three peptide fractions with different molecular sizes were produced for each hydrolysate (i.e., peptide fraction >10 kDa, peptide fraction 3–10 kDa, and peptide fraction <3 kDa). Thus, there were 18 peptide fractions obtained from the ultrafiltration of six hydrolysates using two ultrafiltration membranes of 10 and 3 kDa.

### 3.3. Concentration of Hydrolysates and Peptide Fractions

The protein content (protein concentration) of hydrolysates and peptide fractions were analyzed using the Lowry method. The results are summarized in Table 1.

### 3.4. Antioxidant Activity

The antioxidant activities of the hydrolysates and peptide fractions were tested using the scavenging technique against 2,2-difenil-1-picrylhydrazil (DPPH) radicals. The resulting antioxidant activities are summarized in Figure 2. The antioxidant activities were calculated as specific activities expressed in µmol AAE/g protein. All hydrolysates and peptide fractions showed scavenging activity against DPPH radicals, in the range of 6.17–135.40 µmol AAE/g protein. The two highest antioxidant activities were found in the PBN and PPS fractions <3 kDa, which were 135.40 and 99.78 µmol AAE/g protein, respectively.

### 3.5. Effects of Collagen Types, Enzymes, and Molecular Sizes on Antioxidant Activities

Antioxidant activities were then studied concerning the effects of three factors, namely the collagen types (BSC and PSC), the enzymes used to hydrolyze the collagens (alcalase, neutrase, and savinase), and the molecular sizes (hydrolysate, peptide fraction >10 kDa, peptide fraction 3−10 kDa, and peptide fraction <3 kDa). The effects of these factors on antioxidant activities were studied statistically using IBM SPSS Statistics 20 and expressed as profile plots. The resulting profile plots are shown in Figure 3. Figure 3 shows the effects of the collagen types, the enzymes that hydrolyze collagens to hydrolysates, and the sizes of the molecules on the antioxidant activities produced through the estimated marginal average value.

### 3.6. Amino Acid Sequence

The PBN and PPS fractions <3 kDa were molecules that showed the highest antioxidant activity among all samples with in vitro studies using the DPPH method. The amino acid sequences of the peptides in the PBN and PPS fractions of <3 kDa were analyzed using LC-MS/MS (UHPLC Vanquish Tandem Q Exactive Plus Orbitrap HRMS Thermo Scientific with Pep Map RSLC C18 column, 150 × 0.075 mm, 3 µm, 100 Å). The amino acid sequences of the peptides in the PBN and PPS fractions of <3 kDa are summarized in Table 2.

It can be seen that the PBN and PPS fractions <3 kDa may consist of several peptides, one peptide was for PBN fraction <3 kDa, and 12 peptides were for PPS fraction <3 kDa. All peptides in the PBN and PPS fractions of <3 kDa obtained from the result of LC-MS/MS analysis possessed amino acid residues of 10 to 23 amino acids. The molecular mass varies from 1063.5031 to 2624.1539 Da.

### 3.7. Prediction of Bioactive Properties and Characteristics of Peptides in the PBN and PPS Fractions of <3 kDa

Prediction of bioactive properties of the peptides present in the PBN and PPS fractions <3 kDa has been studied using in silico analysis available on the BIOPEP website. The results of the prediction of bioactive properties for the peptides in the PBN and PPS fractions of <3 kDa can be seen in Table 3.

The characteristics of each of these bioactive peptides were studied using in silico analysis available on the PepDraw website. The results were then studied statistically for the maximum, minimum, first quartile (Q1), second or median quartiles (Q2), and third quartile (Q3) values. The characteristics of the bioactive peptides in the PBN fraction of <3 kDa are summarized in Table 4, and the characteristics of the bioactive peptides in the PPS fraction of <3 kDa are summarized in Table 5 and Table 6.

The number of bioactive peptides characterized by amino acid residues at C-terminal and N-terminal present in the PBN and PPS fractions <3 kDa is summarized in Table 7.

## 4. Discussion

Collagens were successfully extracted from the skin of bigeye tuna (*Thunnus obesus*) using dilute acetic acid and bromelain, as well as using dilute acetic acid and pepsin. The collagen obtained was called bromelain soluble collagen (BSC) for collagen extracted using dilute acetic acid and bromelain, and pepsin soluble collagen (PSC) for collagen extracted using dilute acetic acid and pepsin.

Both bromelain soluble collagen and pepsin soluble collagen provide high collagen yields (data not shown here). Reports on the collagen extraction with bromelain by other researchers are still limited [54]. The high yield of bromelain soluble collagen was revealed in our previous study [46]. Meanwhile, reports on the collagen extraction with pepsin by other researchers have been widely carried out [43,55,56,57,58].

Each protease enzyme works specifically. The bromelain used in this study was stem-bromelain (Sigma-Aldrich), which can be suspended in acetate buffer, pH 4.5 (1 mg/mL), producing a cloudy white suspension. Stem-bromelain is a cysteine endopeptidase with broad specificity for protein cleavage. Stem-bromelain works efficiently on synthetic substrates containing Arg-Arg [59,60]. Pepsin exhibits preferential cleavage of the aromatic residue at one of the peptide bond positions [61]. Allegedly, bromelain and pepsin helped specifically cleave the telopeptide and are cross-linked structures, thereby providing high collagen yields [43,62].

Hydrolysates were obtained from the skin of bigeye tuna (*Thunnus obesus*) through a two-stage hydrolysis process. The first step is the enzymatic hydrolysis stage of the skin into collagen. The second step is the enzymatic hydrolysis stage, which converts the collagen into hydrolysates.

In step two, the collagens (BSC and PSC) were further hydrolyzed using three protease enzymes, namely alcalase, neutrase, and savinase. Alcalase (Sigma-Aldrich) is a serine endoproteinase with broad specificity for native and denatured proteins. Alcalase is active under alkaline conditions, the optimum pH is between 6.5 and 8.5 and has an optimal temperature of 60 °C. Neutrase (Novozymes) is an enzyme with metallo-endoprotease activity that hydrolyzes internal peptide bonds. Savinase (Novozymes) is serine endoproteinase that hydrolyzes internal peptide bonds. Thus, these enzymes will cleave the peptide bonds in the collagens (BSC and PSC) at certain positions, resulting in hydrolysates (collagen hydrolysates) with certain properties.

After obtaining the hydrolysates, the next step is the preparation of the peptide fractions, which were obtained through a semi-purification process of the hydrolysates (collagen hydrolysates) by ultrafiltration technique using 10 and 3 kDa membranes, which separates the peptides based on differences in molecular weight. The choice of membrane size was adjusted to the bioactive target in this study. Small peptide fractions are expected to have bioactive properties with higher activity than the large peptide fractions.

All the hydrolysates and peptide fractions obtained were clear and odorless. These results are in agreement with previous researchers, who also stated that hydrolyzed collagen was neutral and colorless [24].

Oxidation is an essential reaction in all living organisms. The formation of free radicals and other reactive oxygen species (ROS) is unavoidable during the process of oxidative metabolism. These reactive radicals play an important role in signal transduction. However, excessive free radicals can cause destructive effects on living tissues and foodstuffs (degrading food quality and shortening shelf life) [63,64].

Antioxidants work by donating electrons or hydrogen to free radicals. Antioxidants can also stop chain reactions by removing singlet oxygen, deactivating metal catalysts and removing radical initiators or intermediates [65,66,67]. Some artificial antioxidants such as propyl gallate (PG), butylated hydroxyanisole (BHA), and butylated hydroxytoluene (BHT) exhibit strong antioxidant activity and have been used to scavenge free radicals in biological and food systems, but their application is strictly regulated [68,69].

Studies on antioxidant peptides have been reported. These antioxidant peptides were found to have relatively low molecular weight, simple structure, high activity, easy absorption, better stability in different situations, and no harmful immune reactions [70].

In this study, we tested the antioxidant activities of hydrolysates and peptide fractions of collagen from the skin of big eye tuna (*Thunnus obesus*) using a scavenging technique against 2,2-difenil-1-picrylhydrazil (DPPH) radicals. This scavenging technique against DPPH radicals was chosen because it is one of the commonly used techniques in analyzing the antioxidant activity of a compound in vitro. The principle is the DPPH provides maximum absorbance at a wave-length of 517 nm. The presence of antioxidant properties in the hydrolysates and peptide fractions was shown by a decrease in absorbance or a decrease in the intensity of the purple color. In the ascorbic acid solutions of 0.00, 0.31, 0.63, 1.25, 2.50, 5.00, 10.00 mg/L, which were used as positive controls, the color change was clearly visible, as purple turned into yellow. Meanwhile, the color change was not as intensive for the hydrolysates and peptide fractions.

The two highest antioxidant activities were produced by the PBN fraction <3 kDa and PPS fraction <3 kDa (wherein these fractions consisted of peptides <3 kDa). This result is consistent with previous studies, which also found that small peptides gave better results than large peptides [32,33].

In addition, for future research, several test methods can be applied to produce more comprehensive observations of the antioxidant activity study. The selection of the analytical method, the solvent used, the concentration of the test compound, the concentration of the radical compound, and the incubation time also need to be considered.

The antioxidant activities obtained by scavenging technique against DPPH radicals were studied statistically and expressed as profile plots. The factors observed were the collagen types, the enzymes used to hydrolyze collagen into hydrolysates and peptide fractions, and the molecular sizes.

The types of collagens used in this study were bromelain soluble collagen (BSC) and pepsin soluble collagen (PSC). These collagens were obtained from the skin of bigeye tuna (*Thunnus obesus*) using the same method, except for the enzymes used, namely bromelain for BSC and pepsin for PSC. The two collagens gave different effects on antioxidant activity even though they were obtained from the same skin, namely from bigeye tuna skin. The estimated marginal average value of the antioxidant activity of PSC is higher than that of BSC. This shows that in producing antioxidant compounds, overall, PSC gives better results than BSC.

Alcalase, neutrase, and savinase are enzymes used to further hydrolyze the collagen. These enzymes work specifically to produce specific peptides, which in turn affect the antioxidant activity produced. Savinase overall showed the best results compared to neutrase and alcalase in producing antioxidant activity.

Antioxidant activities were also studied as they affected the molecular sizes. The samples studied were the hydrolysate (a mixture of peptides of various molecular sizes), the peptide fraction >10 kDa (i.e., the fraction containing peptides with a molecular weight greater than 10 kDa), the peptide fraction 3–10 kDa (i.e., the fraction containing peptides with a molecular weight between 3 and 10 kDa), and the peptide fraction <3 kDa (i.e., the fraction containing peptides smaller than 3 kDa).

The results of statistical analysis expressed as profile plots showed that the peptide fraction <3 kDa gave the highest antioxidant activity, which was shown from the estimated marginal average value. As we all understand, the smaller the size of the molecule, the less the steric hindrance the compound has, making it easier to react with other compounds. Thus, the peptides in the peptide fraction <3 kDa were easier to react with the DPPH radicals, compared to the peptides in the larger peptide fraction.

Interestingly, the hydrolysates gave a better effect on the resulting antioxidant activities compared to the peptide fractions 3–10 kDa and the peptide fractions >10 kDa. This may be due to the different types of peptides in the hydrolysates.

However, for the semi-purified peptide fraction, the small peptide fraction (the peptide fraction <3 kDa) gave better antioxidant activity than the large peptide fraction (the peptide fractions 3–10 and >10 kDa). In general, antioxidant activity was found to be inversely related to molecular size. For the peptide fraction <3 kDa, the order of antioxidant activity was PBN > PPS > PPN > PPA > PBS > PBA.

A comprehensive study on the effect of these factors on the antioxidant activity produced, especially for the antioxidant activity of hydrolysates and peptide fractions from collagen by-products of the fishery industry, has not been reported. However, our results are consistent with previous studies [32,33] that reported that small peptides provided better antioxidant activity than larger peptides. The peptides in the PBN and PPS fractions of <3 kDa as molecules with the two highest antioxidant activities had a length range of amino acids belonging to the range of bioactive peptides, which has also been reported by previous researchers [17,71].

The combination of in vitro and in silico methods can increase the discovery of bioactive peptides for further applications in food and health [36]. This combination of methods quickens, simplifies, and reduces research costs. The amino acid sequences of the peptides in the PBN and PPS fractions of <3 kDa were analyzed by LC-MS/MS (in vitro study with the DPPH method), then used for in silico analysis to predict their bioactive properties and to study their characteristics.

In the PBN fractions of <3 kDa, one peptide with a residue length of 23 amino acids was identified according to the results of the LC-MS/MS analysis. From the in silico results with the BIOPEP program, 16 dipeptides from this peptide sequence were indicated to possess bioactive properties. The sixteen dipeptides were PW, FP, LF, VG, GR, DA, GS, HG, DY, YV, AD, HF, SL, WC, TV, VH. These 16 dipeptides were grouped as antidiabetic peptides 81.25%, antihypertensive peptides 56.25% and antioxidant peptides 6.25%. Among these peptides, there were those predicted as multifunctional peptides, namely 25% as antidiabetic and antihypertensive (FP, VG, DA, YV); and 6.25% as antidiabetic and antioxidant (PW). The identified antidiabetic peptides were alpha-glucosidase inhibitors 6.25%, dipeptidyl peptidase III inhibitors 12.50%, and dipeptidyl peptidase IV inhibitors 62.50%. Antihypertensive peptides were identified as ACE inhibitors.

In the PPS fractions of <3 kDa, 12 peptides with residue lengths of 10–20 amino acids were identified according to the results of LC-MS/MS analysis. From the in silico results with the BIOPEP program, there were 83 peptides from the peptide sequences that have bioactive properties. The eighty-three peptides were KD, IY, GPP, LK, GAA, PGP, PP, EA, AD, WM, SM, PF, YG, GE, PR, GPRG, GPRGP, GPRGPP, GPR, GP, PG, GPGG, LSP, IG, GS, NY, MW, VF, FP, NK, GEP, KG, GT, TP, RG, ILP, IL, AP, GA, VG, GL, MG, QG, GD, EV, PQ, VGP, DM, MGP, AG, GG, TG, GRP, RP, GR, NG, DG, GPA, AA, AGP, QGP, GV, GAGP, SP, SL, WC, DR, PN, SW, YD, KK, EP, NP, PPG, PS, LP, DN, NV, PK, QL, VI, PA, GPAG. These 83 peptides were grouped as 73.49% antidiabetic peptides, 61.45% antihypertensive peptides, 6.02% antioxidant peptides and 15.66% peptides with other bioactive properties (i.e., chemotactic, antithrombotic, antiamnestic). Among these peptides, there were those that function as multifunctional peptides, namely 34.94% as antihypertensive and antidiabetic (PP, EA, WM, YG, GE, PR, NY, MW, VF, FP, KG, TP, RG, AP, GA, VG, GL, MG, QG, EV, PQ, AG, GG, TG, RP, NG, GPA, AA, GV); 2.41% as antioxidant and antihypertensive (IY, GPP); 1.20% as chemotactic, antithrombotic, and antiamnestic (PGP); 2.41% as antithrombotic, antihypertensive, antidiabetic, and antiamnestic (GP, PG); 1.20% as antithrombotic, and antiamnestic (GPGG). The identified antidiabetic peptides had activity as alpha-glucosidase inhibitors in the amount of 3.61%, dipeptidyl peptidase III inhibitors in the amount of 7.23%, and dipeptidyl peptidase IV inhibitors in the amount of 62.65%. Antihypertensive peptides were identified as ACE inhibitors.

Bioactive peptides in the PBN and PPS fractions of <3 kDa, namely antidiabetic peptides, antihypertensive peptides, antioxidant peptides, and peptides with other bioactive properties, generally have the N terminal of a non-polar (hydrophobic) amino acid. This finding is in agreement with previous researchers, who also linked bioactive properties with hydrophobicity [72]. The bioactive peptides in the PBN and PPS fractions of <3 kDa appeared as sequences of 2–6 amino acids with a molecular weight of 132.05–579.31. This range of amino acid residue lengths matches the range of bioactive peptide residue as reported by previous investigators [17,71,73].

## 5. Conclusions

Research on bioactive peptides from the by-product of the tuna fishery industry has been successfully carried out using an integrated method. This combination research is considered cheaper, easier and faster in the research for bioactive peptides.

Further hydrolysis using alcalase, neutrase and savinase, as well as their fractionation, showed the best antioxidants in the PBN fraction (Bromelain-Neutrase Peptide Fraction) of <3 kDa and PPS fraction (Pepsin-Savinase Peptide Fraction) of <3 kDa. The collagen types, enzymes, and molecular sizes affect the antioxidant activities produced. The in silico test showed the peptides in the PBN and PPS fractions of <3 kDa both had antidiabetic, antihypertensive, and antioxidant activities. Several bioactive peptides in the PBN and PPS fractions of <3 kDa were found to have multifunctional properties. The bioactive peptides in the PBN and PPS fractions of <3 kDa generally have the N-terminal of a non-polar (hydrophobic) amino acid.

## Figures and Tables

**Figure 1 foods-10-02739-f001:**
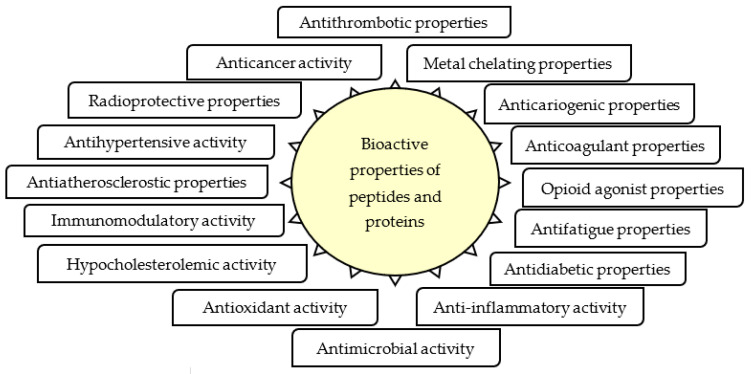
Array of bioactive properties of peptides and proteins [3,9,10,11,12,13,14,15,16].

**Figure 2 foods-10-02739-f002:**
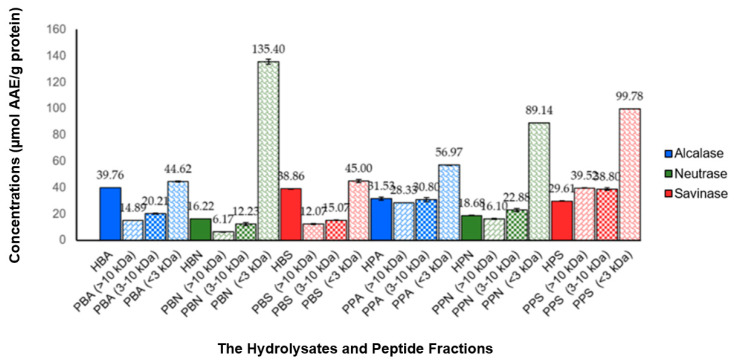
Antioxidant activities of hydrolysates and peptide fractions tested using the scavenging technique against DPPH radicals. Antioxidant activities were expressed in µmol AAE/g protein. All values were given as mean ± SD, *n* = 2. The values of antioxidant activities for each hydrolysate and its peptide fractions showed a significant difference with the one-way ANOVA test (*p* < 0.05).

**Figure 3 foods-10-02739-f003:**
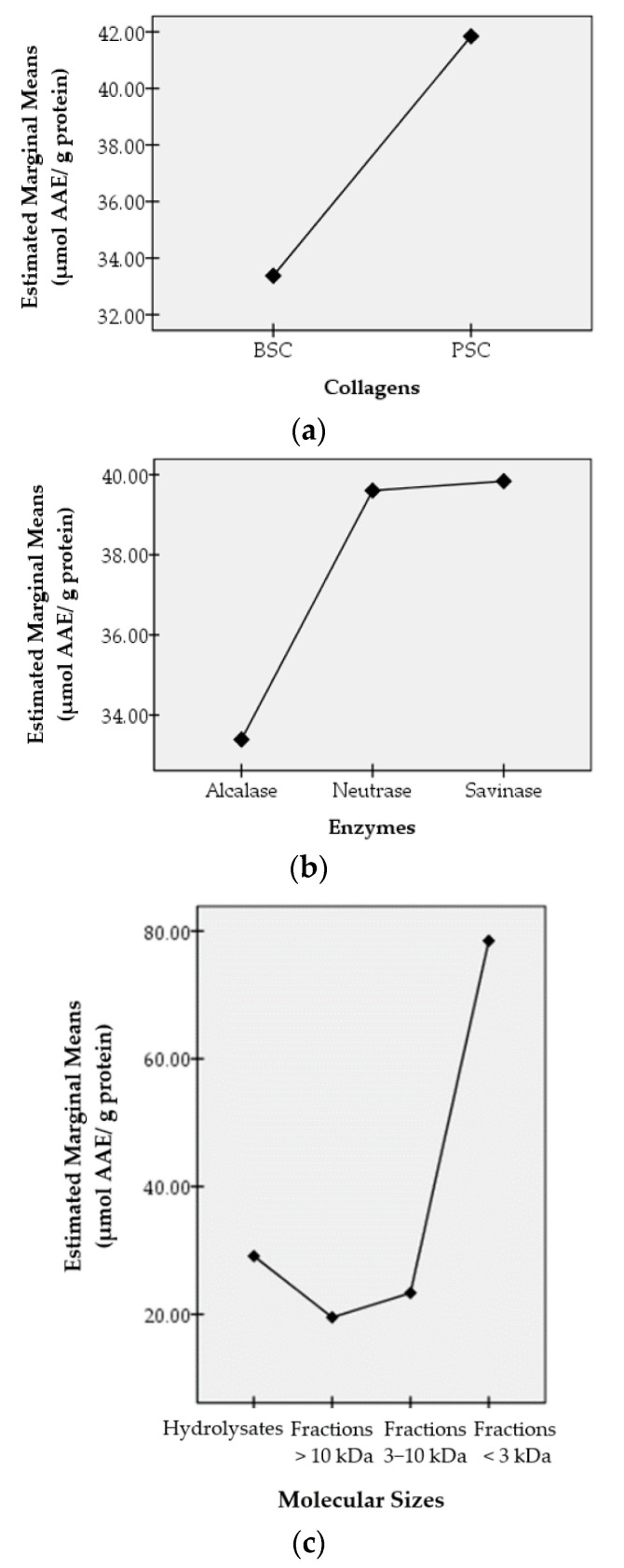
Profile plots of factors influencing the antioxidant activity of hydrolysates and peptide fractions of bigeye tuna skin collagen. The X-axis shows the type of collagen (**a**), the type of enzyme (**b**), and the size of the molecule (**c**). The Y-axis shows the estimated marginal average value (in µmol AAE/g protein).

**Table 1 foods-10-02739-t001:** Protein content (mg/mL) of hydrolysates and peptide fractions analyzed by the Lowry method.

Sample	Content	Sample	Content	Sample	Content
HBA	1.03 ± 0.02	HBN	1.99 ± 0.00	HBS	1.11 ± 0.09
PBA > 10 kDa	2.12 ± 0.00	PBN > 10 kDa	4.24 ± 0.08	PBS > 10 kDa	1.75 ± 0.06
PBA 3–10 kDa	1.71 ± 0.04	PBN 3–10 kDa	1.95 ± 0.04	PBS 3–10 kDa	1.69 ± 0.03
PBA < 3 kDa	0.75 ± 0.00	PBN < 3 kDa	0.11 ± 0.00	PBS < 3 kDa	0.50 ± 0.00
HPA	1.23 ± 0.03	HPN	2.04 ± 0.07	HPS	1.34 ± 0.05
PPA > 10 kDa	0.97 ± 0.02	PPN > 10 kDa	2.52 ± 0.00	PPS > 10 kDa	1.14 ± 0.00
PPA 3–10 kDa	0.93 ± 0.04	PPN 3–10 kDa	1.25 ± 0.00	PPS 3–10 kDa	1.02 ± 0.04
PPA < 3 kDa	0.44 ± 0.00	PPN < 3 kDa	0.22 ± 0.00	PPS < 3 kDa	0.23 ± 0.02

All values were given as mean ± standard deviation, *n* = 2. The values of protein contents between the hydrolysates and their peptide fractions showed significant differences with the one-way ANOVA test (*p* < 0.05). HBA, HBN, and HBS are BSC hydrolysates. HPA, HPN, and HPS are PSC hydrolysates. PBA, PBN, and PBS are the ultrafiltration fractions of HBA, HBN, and HBS, respectively. PPA, PPN, and PPS are the ultrafiltration fractions of HPA, HPN, and HPS, respectively. The index value indicates the molecular size of the peptide in fractions, namely: >10, 3–10, <3 kDa.

**Table 2 foods-10-02739-t002:** Amino acid sequences of peptides in the PBN and PPS fractions of <3 kDa.

Peptide Fraction	Amino Acid Sequence	Acc. Number	Length	Mass
PBN < 3 kDa	HFDcTVHGSLFPWcSLDADYVGR	P02784	23	2624.1539
PPS < 3 kDa	IGSmWmSWcSLSPNYDKDR	P02784	19	2274.9827
	cVFPFIYGNKK	P04557	11	1314.6773
	KGEPGTPGNPGPRGPPGSPS	H2M082	20	1841.8992
	DEQLSEDNVILPK	P04557	13	1498.7489
	PGEPGTPGEPGAP	A0A1G7X1M5	13	1161.5284
	cVFPFIYGNK	P04557	10	1186.5826
	GPRGPPGEPGPPGSPG	A0A3P6HM70	16	1411.6823
	EVGPQGLKGDMGPQ	A0A3B3BHI3	14	1411.6743
	GGADTGEVGPRDTGEAG	A0A498LDH1	17	1544.6680
	PGEPGPPGPNGEDGRP	A0A0M3JYC0	16	1528.6884
	GPAGPGGAAGAQGP	A0A536PFB4	14	1063.5031
	PGAGPGGAGPGVGGG	A0A368EWB0	15	1063.5031

The lengths and masses of the peptides in the PBN and PPS fractions of <3 kDa were obtained from the PepDraw website.

**Table 3 foods-10-02739-t003:** Prediction of bioactive properties of peptides in the PBN and PPS fractions of < 3 kDa.

Peptide Fractions	PBN < 3 kDa	PPS < 3 kDa											
Amino acid sequence	HFDcTVHGSLFPWcSLDADYVGR	IGSmWmSWcSLSPNYDKDR	cVFPFIYGNKK	KGEPGTPGNPGPRGPPGSPS	DEQLSEDNVILPK	PGEPGTPGEPGAP	cVFPFIYGNK	GPRGPPGEPGPPGSPG	EVGPQGLKGDMGPQ	GGADTGEVGPRDTGEAG	PGEPGPPGPNGEDGRP	GPAGPGGAAGAQGP	PGAGPGGAGPGVGGG
Accession number	P02784	P02784	P04557	H2M082	P04557	A0A1G7X1M5	P04557	A0A3P6HM70	A0A3B3BHI3	A0A498LDH1	A0A0M3JYC0	A0A536PFB4	A0A368EWB0
Antioxidant	PW	KD	IY	GPP			IY	GPP	LK		GPP	GAA	
ACE inhibitors	FP, LF, VG, GR, DA, GS, HG, DY, YV	LSP, IG, GS, NY, WM, MW	IY, VF, FP, YG, NK	PR, GP, GEP, KG, GS, GT, GE, PG, GPP, PP, TP, RG	ILP, IL	GEP, AP, GA, GT, GE, PG, TP	IY, VF, FP, YG, NK	PR, GP, GEP, GS, GE, PG, GPP, PP, RG	GP, VG, GL, KG, MG, QG, GD, EV, PQ, VGP, DM, MGP	PR, GP, VG, GA, AG, GE, GG, TG, EA, EV, VGP	GRP, GP, GEP, RP, GR, GE, NG, PG, DG, GPP, PP	GPA, GP, AA, GA, AG, GG, QG, PG, AGP, QGP	GP, VG, GA, AG, GV, GG, PG, AGP, GAGP
Inhibitor				PGP				PGP			PGP, PPGP		
Chemotactic				PGP				PGP			PGP		
Alpha-glucosidase inhibitor	AD			PP				PP		EA, AD	PP		
Dipeptidyl peptidase III inhibitor	DA, HF	WM, SM	PF, YG	GE, PR		GE	PF, YG	GE, PR		GE, PR	GE		
Dipeptidyl peptidase IV inhibitor	FP, SL, WC, AD, HF, PW, TV, VG, VH, YV	SP, SL, WM, WC, MW, DR, NY, PN, SW, YD	FP, KK, PF, VF, YG	GP, PP, TP, SP, EP, NP, PPG, GE, KG, PG, PS, RG	LP, DN, IL, NV, PK, QL, VI	AP, TP, GA, EP, GE, PG	FP, PF, VF, YG	GP, PP, SP, EP, PPG, GE, PG, RG	GP, GL, EV, KG, MG, PQ, QG, VG	GP, GA, AD, AG, EV, GE, GG, TG, VG	GP, PP, RP, EP, PPG, GE, NG, PG, PN	GP, PA, GPA, GA, AA, AG, GG, PG, QG, GPAG	GP, GA, AG, GG, GV, PG, VG
Antithrombotic				GPRG, GPRGP, GPRGPP, GPR, GP, PGP, PG		PG		GPRG, GPRGP, GPRGPP, GPR, GP, PGP, PG	GP	GPR, GP	GP, PGP, PG	GP, PG, GPGG	GP, PG, GPGG
Antiamnestic				PGP, PG, GP		PG		PGP, PG, GP	GP	GP	PGP, PG, GP	GPGG, PG, GP	GPGG, PG, GP

Prediction of bioactive properties of peptides in the PBN and PPS fractions of <3 kDa was studied using the in silico method found at the BIOPEP website. The letters in columns 2–14 are the symbols for the amino acids in the peptide.

**Table 4 foods-10-02739-t004:** The characteristics of the bioactive peptides in the PBN fraction of <3 kDa.

Characteristic	Statistics	Antioxidant	ACE Inhibitor	Chemotactic	Alpha-Glucosidase Inhibitor
Amino acid sequence	-	PW	FP, LF, VG, GR, DA, GS, HG, DY, YV	AD	DA, HF
Percentage of bioactivity	-	6.25	56.25	6.25	12.50
Sequence length	Max	2.00	2.00	2.00	2.00
	Min	2.00	2.00	2.00	2.00
	Q_1_	2.00	2.00	2.00	2.00
	Q_2_	2.00	2.00	2.00	2.00
	Q_3_	2.00	2.00	2.00	2.00
Molecular weight	Max	301.14	296.10	204.07	302.14
	Min	301.14	162.06	204.07	204.07
	Q_1_	301.14	196.58	204.07	228.59
	Q_2_	301.14	246.63	204.07	253.11
	Q_3_	301.14	278.66	204.07	277.62
Isoelectric point	Max	5.21	10.73	3.13	7.68
	Min	5.21	2.95	3.13	2.95
	Q_1_	5.21	5.45	3.13	4.13
	Q_2_	5.21	5.51	3.13	5.32
	Q_3_	5.21	5.64	3.13	6.50
Net charge	Max	0.00	1.00	−1.00	0.00
	Min	0.00	−1.00	−1.00	−1.00
	Q_1_	0.00	0.00	−1.00	−0.75
	Q_2_	0.00	0.00	−1.00	−0.50
	Q_3_	0.00	0.00	−1.00	−0.25
Hydrophobicity (Kcal/mol)	Max	5.95	12.04	12.04	12.04
	Min	5.95	4.94	12.04	8.52
	Q_1_	5.95	6.73	12.04	9.40
	Q_2_	5.95	9.51	12.04	10.28
	Q_3_	5.95	10.86	12.04	11.16

Min, max, Q1, Q2, and Q3 are the minimum, maximum, first quartile, second quartile (median) and third quartiles, respectively. All characteristics were studied using the PepDraw website.

**Table 5 foods-10-02739-t005:** The characteristics of the bioactive peptides in the PPS fraction of <3 kDa.

Characteristic	Statistics	Antioxidant	ACE Inhibitor	Chemotactic	Alpha-Glucosidase Inhibitor
Amino acid sequence	-	KD, IY, GPP, LK, GAA	LSP, IG, GS, NY, WM, MW, IY, VF, FP, YG, NK, PR, GP, GEP, KG, GT, GE, PG, GPP, PP, TP, RG, ILP, IL, AP, GA, VG, GL, MG, QG, GD, EV, PQ, VGP, DM, MGP, AG, GG, TG, EA, GRP, RP, GR, NG, DG, GPA, AA, AGP, QGP, GV, GAGP	PGP	PP, EA, AD
Percentage of bioactivity	-	6.02	61.45	1.20	3.61
Sequence length	Max	3.00	4.00	3.00	2.00
	Min	2.00	2.00	3.00	2.00
	Q_1_	2.00	2.00	3.00	2.00
	Q_2_	2.00	2.00	3.00	2.00
	Q_3_	3.00	2.00	3.00	2.00
Molecular weight	Max	294.16	341.23	269.14	204.07
	Min	217.11	146.07	269.14	204.07
	Q_1_	259.19	196.59	269.14	204.07
	Q_2_	261.13	244.18	269.14	204.07
	Q_3_	269.14	282.66	269.14	204.07
Isoelectric point	Max	9.80	11.56	5.25	5.25
	Min	5.48	2.95	5.25	3.09
	Q1	5.61	5.38	5.25	3.11
	Q2	5.65	5.57	5.25	3.13
	Q3	6.77	5.65	5.25	4.19
Net charge	Max	1.00	1.00	0.00	0.00
	Min	0.00	−1.00	0.00	−1.00
	Q_1_	0.00	0.00	0.00	−1.00
	Q_2_	0.00	0.00	0.00	−1.00
	Q_3_	0.00	0.00	0.00	−0.50
Hydrophobicity (Kcal/mol)	Max	14.34	12.82	9.33	12.04
	Min	6.07	5.14	9.33	8.18
	Q_1_	9.33	8.32	9.33	10.11
	Q_2_	9.45	9.30	9.33	12.03
	Q_3_	10.05	10.52	9.33	12.04

Min, max, Q1, Q2, and Q3 are the minimum, maximum, first quartile, second quartile (median) and third quartiles, respectively. All characteristics were studied using the PepDraw website.

**Table 6 foods-10-02739-t006:** The characteristics of the bioactive peptides in the PPS fraction of <3 kDa.

Characteristic	Statistics	Dipeptidyl Peptidase III Inhibitor	Dipeptidyl Peptidase IV Inhibitor	Antithrombotic	Antiamnestic
Amino acid sequence	-	WM, SM, PF, YG, GE, PR	SP, SL, WM, WC, MW, DR, NY, PN, SW, YD, FP, KK, PF, VF, YG, GP, PP, TP, EP, NP, PPG, GE, KG, PG, PS, RG, LP, DN, IL, NV, PK, QL, VI, AP, GA, GL, EV, MG, PQ, QG, VG, AD, AG, GG, TG, RP, NG, PA, GPA, AA, GPAG, GV	GPRG, GPRGP, GPRGPP, GPR, GP, PGP, PG, GPGG	PGP, PG, GP, GPGG
Percentage of bioactivity	-	7.23	62.65	9.64	4.82
Sequence length	Max	2.00	4.00	6.00	4.00
	Min	2.00	2.00	2.00	2.00
	Q_1_	2.00	2.00	2.75	2.00
	Q_2_	2.00	2.00	3.50	2.50
	Q_3_	2.00	2.00	4.25	3.25
Molecular weight	Max	271.16	335.13	579.31	286.13
	Min	204.07	132.05	172.08	172.08
	Q_1_	220.08	216.11	211.11	172.08
	Q_2_	236.08	244.11	356.70	220.61
	Q_3_	253.62	267.66	458.00	273.38
Isoelectric point	Max	10.73	11.56	11.56	5.65
	Min	3.21	2.95	5.23	5.23
	Q1	5.24	5.25	5.51	5.25
	Q2	5.43	5.45	8.19	5.43
	Q3	5.57	5.61	11.24	5.61
Net charge	Max	1.00	2.00	1.00	0.00
	Min	−1.00	−1.00	0.00	0.00
	Q_1_	0.00	0.00	0.00	0.00
	Q_2_	0.00	0.00	0.50	0.00
	Q_3_	0.00	0.00	1.00	0.00
Hydrophobicity (Kcal/mol)	Max	12.68	13.50	12.43	11.49
	Min	5.14	5.14	9.19	9.19
	Q_1_	6.67	7.98	9.30	9.19
	Q_2_	8.02	8.85	11.25	9.26
	Q_3_	9.47	9.98	12.19	9.87

Min, max, Q1, Q2, and Q3 are the minimum, maximum, first quartile, second quartile (median) and third quartiles, respectively. All characteristics were studied using the PepDraw website.

**Table 7 foods-10-02739-t007:** Number of bioactive peptides characterized by amino acid residues at C-terminal and N-terminal present in the PBN and PPS fractions <3 kDa.

Peptide Fractions	Antidiabetic Peptides		Antihypertensive Peptides		Antioxidant Peptides		Other Bioactive Peptides	
	C-Terminal	N-Terminal	C-Terminal	N-Terminal	C-Terminal	N-Terminal	C-Terminal	N-Terminal
PBN < 3 kDa	6 non polar (hydrophobic), 1 positively charged, 1 negatively charged, 3 polar uncharged	7 non polar (hydrophobic), 1 positively charged, 1 negatively charged, 2 polar uncharged	3 non polar (hydrophobic), 1 positively charged, 2 negatively charged, 3 polar uncharged	4 non polar (hydrophobic), 1 positively charged, 4 polar uncharged	1 non polar (hydrophobic)	1 non polar (hydrophobic)	-	-
PPS < 3 kDa	24 non polar (hydrophobic), 4 positively charged, 5 negatively charged, 22 polar uncharged	29 non polar (hydrophobic), 4 positively charged, 3 negatively charged, 19 polar uncharged	21 non polar (hydrophobic), 3 positively charged, 4 negatively charged, 23 polar uncharged	28 non polar (hydrophobic), 3 positively charged, 2 negatively charged, 18 polar uncharged	2 non polar (hydrophobic), 1 positively charged, 2 polar uncharged	2 non polar (hydrophobic), 1 positively charged, 1 negatively charged, 1 polar uncharged	2 non polar (hydrophobic), 6 polar uncharged	4 non polar (hydrophobic), 1 positively charged, 3 polar uncharged

## Data Availability

The datasets generated for this study are available on request to the corresponding author.

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
