# Peer review of "The Bioactivity Prediction of Peptides from Tuna Skin Collagen Using Integrated Method Combining In Vitro and In Silico"

_foods, 2021, doi:10.3390/foods10112739_

Round 1
Reviewer 1 Report
Dear Authors,
The manuscript contains satisfactory information about of the bioactive peptides from the by-product of the tuna fishery industry, and has been successfully carried out using an integrated methods, i.e. enzyme extraction, enzymatic hydrolysis, ultrafiltration, and most important the combination of data research in order to predict the various bioactivities of peptide fractions taking into account the determined antioxidant activity.
All mentioned interaction analyses, presented results and the quality of cited literature are satisfied. The paper is scientifically sound, it is proper written and well organized, and it can be understood by non-specialists.
According mentioned above information, the reviewer recommends below: Recommending minor revision. The partial sections in the text of manuscript have to be clarified or fixed/reorganized which could improve the quality of manuscript. The corrections or comments are detailed written in the file labeled as “Reviewer comments”, that will be attached.
Kind regards.

Author Response
Thank you for your review.
Best Regards
Prof. Dr. Ir. Maggy T Suhartono
Corresponding author

Reviewer 2 Report
The study highlights the hydrolysates and peptide fractions of bigeye tuna (Thunnus obesus) skin collagen. Amino acid sequencing by LC-MS / MS and proteomic analysis in silico provide important details regarding the bioactivity of peptides in tuna skin collagen.
Please argue why it is necessary to express the antioxidant activity through two units of measurement. What additional information does it bring us? The information is redundant, repetitive, one of the options should be given up.
In each graph (figure 2 and figure 3) on the OY axis the unit of measurement must be entered.
In vitro, in vivo, in silico it must be written in italic throughout the manuscript.
We suggest the analysis of the antimicrobial activity of the peptide fractions obtained.

Author Response
Thank you for your review. Based on your advice, we have revised our manuscript as follows:
- Thank you for your comment. We agree with you to use one unit of measurement to express antioxidant activity. Revision is in the revised manuscript (line 168, 262, 264, 265, 268, 282-284 in the revised manuscript).
- We have added the unit of measurement in Figure 2 and Figure 3.
- We have revised in vitro, in vivo and in silico writing with italics throughout the manuscript.
- We previously tested antimicrobial activity for all of our samples (hydrolysates and peptides). The results obtained were all negative (not showing antimicrobial activity). Antimicrobial activity tests have been carried out on coli, B.Subtilis, S.Aureus, C.albicans. Because the results were all negative, and based on the in silico test, there were no results antimicrobial activities, so we did not include these results in the manuscript.
Best Regards,
Prof. Dr. Ir. Maggy T Suhartono
(Corresponding author)

Round 2
Reviewer 1 Report
Dear Authors,
The manuscript has been successfully corrected in accordance with the Reviewer’s proposed suggestions and questions asked. Thanks for all the answers. With detailed additions, the manuscript has been significantly improved and as such is ready for further process of publication in the journal Foods.
Kind regards.